# Assessments of and Attitudes towards Specialist Teleconsultations among Nephrology and Posttransplant Outpatients during the COVID-19 Pandemic

**DOI:** 10.3390/healthcare11202737

**Published:** 2023-10-14

**Authors:** Krzysztof Edyko, Paweł Edyko, Maja Nowicka, Ilona Kurnatowska

**Affiliations:** 1Student Scientific Society Affiliated with the Department of Internal Medicine and Transplant Nephrology, Chair of Pulmonology, Rheumatology, and Clinical Immunology, Medical University of Łódź, Tadeusza Kościuszki 4, 90-419 Łódź, Poland; 2Department of Internal Medicine and Transplant Nephrology, Chair of Pulmonology, Rheumatology, and Clinical Immunology, Medical University of Łódź, Kopcińskiego 22, 90-153 Łódź, Poland

**Keywords:** patient satisfaction, telemedicine, digital health, COVID-19 pandemic, nephrology, transplantology

## Abstract

In Poland, teleconsultations (TCs) were not legally regulated or even conducted until the COVID-19 pandemic, which necessitated their abrupt implementation and posed a challenge to patients and doctors. The aim of this study was to assess the quality of TCs and the satisfaction with this mode of consultation among nephrology and kidney transplant outpatients with a high risk of severe courses of SARS-CoV-2 infection. A self-designed questionnaire regarding patients’ demographics; digital fluency; and participation in, satisfaction with, and attitude towards TCs was distributed among patients in the nephrology and posttransplant outpatient clinics at two hospitals in central Poland. The questionnaires were completed by 294 adult patients, of whom 72.1% (*n* = 212) had participated in TCs at one of the abovementioned clinics. Almost all (96.7%) of the TCs were conducted via phone, and in 94.8% of cases, they fulfilled the purpose of the consultation. The most commonly reported advantages were not having to leave home and the reduced risk of infection. Only a few patients felt that TCs offer no advantages. The patients’ profiles and demographic data had no significant effect on their assessments of teleconsultations. Despite the overall positive rating given to TCs, patients unhesitatingly indicated that a face-to-face visit would be a preferable way to contact a specialist.

## 1. Introduction

Severe acute respiratory syndrome coronavirus 2 (SARS-CoV-2) was identified in Wuhan, China, at the end of 2019 [1]. Due to its high level of virulence, the virus spread worldwide, and on March 11 2020, the World Health Organization (WHO) declared the COVID-19 outbreak a pandemic [2]. The rapidly increasing number of confirmed COVID-19 cases resulted in the transformation of numerous hospital units into infectious disease wards; many doctors were quarantined or, in case of infection, isolated, which led to a healthcare breakdown [3,4]. 

In order to prioritize public health, reduce the spread of infection, and prevent an overload of the healthcare system, numerous restrictions were introduced—from the wearing of masks and social distancing to the prohibition of public gatherings and the closure of numerous services [5]. As people were strongly advised to stay at home, the development of remote services accelerated.

Telemedicine, literally “healing at a distance”—a term coined in the 1970s [6]—in practice means conducting consultations between clinicians and patients through different kinds of electronic devices. A teleconsultation is defined as a synchronous or asynchronous consultation that uses information and communication technology to overcome geographical and functional distances [7]. In Poland, even before the pandemic, an increasing emphasis had been put on digitalization in medical services; this resulted in the implementation of the Patient Internet Account (PIA), a system that enables patients to manage their prescriptions and referrals, and have an insight into their past medical history. However, unlike other countries (e.g., Australia, China, and the USA [8,9,10]), in Poland, telemedicine had not been used and remote visits had not been regulated by law until 2019. On November 4 2019, the possibility of conducting teleconsultations (TCs) in primary healthcare units was realized following a regulation enacted by the Minister of Health [11]; during the pandemic, this was extended to specialist care units. All outpatient clinics were asked to replace traditional ambulatory visits with TCs within a matter of weeks.

The pandemic prompted a significant increase in research on the usefulness of remote consultations [12,13]. The main limitation of telemedicine is that it does not permit the clinician to conduct a complete physical examination, and this may result in the failure to observe abnormalities that exert a negative impact on patients’ health. Other drawbacks include potential technical problems, such as connection disruptions and difficulties in understanding the doctor’s orders properly, which especially effects elderly people; those in worse general condition; and those with visual or hearing impairments, in whose cases the assistance of an accompanying person seems necessary. The merits of TCs are primarily associated with minimizing human contact, which certainly reduces the possibility of infection, both while traveling to a clinic and being in a waiting room, while also saving time and reducing costs.

Numerous researchers and healthcare providers, including Polish National Healthcare, have assessed the numbers, duration and evaluation scores of TCs performed in primary healthcare centers [14,15]. However, TCs that are performed in specialist medical units have been analyzed to a far lesser degree. Patients with chronic kidney disease (CKD), especially kidney transplant recipients (KTx), belong to a group at high risk of complications and severe courses of COVID-19 [16]. Furthermore, CKD patients are characterized by many comorbidities, including diabetes mellitus, hypertension, and obesity [17]. Immunosuppressive therapy can also increase the risk of severe courses of SARS-CoV-2 infection [18]. Until a COVID-19 vaccination became available, social isolation had been the only efficient protection against the infection; it was recommended by the WHO and followed by scientific societies, including nephrology and transplant centers [19]. Traditional face-to-face visits in outpatient clinics posed a great risk of infection due to the aggregation of patients from different environments. This threat rapidly led to the introduction of TCs in specialist healthcare settings, including nephrology and posttransplant outpatient clinics. In Poland, TCs were mainly conducted via phone [20]. In Lodz Voivodeship, which is inhabited by 2.45 million people, there are 15,761 patients who remain under the care of nephrologists [21]. Following the WHO’s recommendation [5], from March 2020 to the end of 2021, their traditional ambulatory visits were frequently replaced by TCs.

The aim of our study was to assess the level of satisfaction and attitude towards TCs, as well as their prevalence, among patients in nephrology and posttransplant outpatient clinics and to determine whether TCs were only a helpful tool to control the COVID-19 pandemic or whether, in the opinions of patients, they should continue be used to partially replace specialist in-person visits in the future.

## 2. Materials and Methods

### 2.1. Population

This study was performed in two nephrology clinics and two transplantology outpatient clinics at two hospitals (one university and one municipal) in Central Poland. In 2021, a total of 3729 patients (3162 nephrology patients and 567 posttransplant patients) had at least one visit to one of the studied clinics. This questionnaire-based study was conducted between January and June 2022. All adult patients who remained under the care of nine nephrologists and five transplantologists at the centers involved in the study were asked to complete the survey during routine, ambulatory appointments. The patients filled in the questionnaires in the presence of a research team member who could help with potential queries and provide reading assistance if necessary.

### 2.2. Questionnaire

A self-designed survey was created by an expert group (5 nephrology and transplantology specialists) and validated in 25 patients from nephrology or posttransplant outpatient clinics currently hospitalized in the nephrology ward. The questionnaire consisted of 32 multiple choice questions and 1 open question related to patients’ demographics; digital literacy; participation in and attitude towards TCs; and the level of satisfaction, which was further assessed using the Net Promoter Score (NPS) (Appendix A).

The NPS, introduced in 2003 by Fred Reichheld [22], is a simple way to assess patients’ satisfaction in surveys. It is based on a single survey question: would you recommend this company/service to a friend? [22]. Respondents mark their willingness to recommend a service to a family member or a friend on a scale ranging from 0 (“not at all likely”) to 10 (“extremely likely”). According to the NPS, the respondents who mark 9 or 10 are denoted as “promoters”, 7 or 8 are treated as “passives”, and those who tick 6 or less are “detractors”. The NPS is then calculated as the percentage of “promoters” minus the percentage of “detractors”, and its value ranges from −100% to +100% [23]. 

### 2.3. Statistical Analysis

All the data and questionnaires were anonymized prior to the analysis, which was performed with Statistica 13.3 licensed to Medical University of Lodz. 

Nominal data are presented as numbers with percentages and were compared with the Chi2 test (if the number of cases in each subgroup exceeded 15), the Chi2 test with Yates correction (if the number of cases was between 5 and 15 in at least one subgroup) or with Fisher’s exact test (if <5 cases were present in any subgroup). Correlations between continuous and ordinal variables were tested with the Pearson or Spearman rank correlation test, respectively. *P* values lower than 0.05 were considered statistically significant. All analyses were performed with Microsoft Excel (Microsoft, Redmond, WA, USA) or Statistica 13.0 software (Dell, Round Rock, TX, USA).

## 3. Results

### 3.1. Population

We asked a total of 327 people to fulfil the questionnaire and achieved an 89.9% response rate (*n* = 294). The demographic data of all respondents are presented in Table 1. Only the data of the patients who participated in at least one TC in the posttransplant or nephrology outpatient clinics (the TC group) were further analyzed. In the TC group, we observed a slight advantage in the number of males (*n* = 114, 53.8%). Pensioners were the most numerous group (*n* = 123, 58%), and the patients living in rural areas constituted a percentage similar to that of the inhabitants of cities of more than 500,000 residents.

### 3.2. Teleconsultations

In our study, 72.1% (*n* = 212) of 294 patients participated in TCs. Their characteristics are presented in Table 2. Among the 27.9% (*n* = 82) who did not participate in TCs at one of our clinics, 30.5% (*n* = 25) had a remote visit at another specialist outpatient clinic, 25.6% (*n* = 21) used TCs only with their family doctor, and 43.9% (*n* = 36) did not use TCs at all.

### 3.3. Type of TCs

Almost all specialist TCs were conducted via phone (*n* = 205, 96.7%), with a few exceptions when conducted by email (*n* = 2, 0.94%). None of the patients had a visit via videoconference. The most common aim of a TC was a routine follow-up appointment (*n* = 130, 61.3%) to discuss laboratory results (*n* = 100, 47.2%) and obtain prescriptions (*n* = 99, 46.7%). Rarely did the patients consult new complaints remotely (*n* = 52, 24.5%). A total 77.4% (*n* = 164) of the TC group reported complete understanding of their doctors’ orders; 18.9% (*n* = 40) understood most of them; and for 1.4% of patients (n = 3), the doctor’s recommendations were incomprehensible. Only 5.7% (*n* = 12) needed assistance from a relative during the TC, and those were not only seniors but also two patients aged 21 and 39 with mental disability.

### 3.4. Attitude towards Telemedicine

In patients’ opinion, 94.8% (*n* = 201) of their TCs fulfilled the purpose of the consultation.

When asked about advantages and disadvantages (Table 3). The TC group reported the advantages of a lower risk of infection (*n* = 115, 54.2%) almost as often as the convenience of receiving medical counsel at home (*n* = 130, 61.3%). Among other advantages, 0.9% (*n* = 2) of the respondents listed a time-saving factor and significant convenience for disabled people. Only 9.9% (*n* = 21) saw no benefits of telemedicine. The most often reported downside of a TC was that it did not allow for a physical examination (*n* = 125, 59.0%); 0.9% (*n* = 2) mentioned the absence of eye contact during a remote visit; and 22.6% (*n* = 48) of the patients found no drawbacks in TCs.

A total 85.3% (*n* = 181) of the TC group rated specialist TCs positively and 64.7% (*n* = 137) were willing to use telemedicine in the future. 

Nevertheless, the calculated NPS had a negative value and stood at −14.56.

Except for obtaining prescriptions, where we observed only a minor difference, in the rest of cases there was an overwhelming difference in favor of traditional, face-to-face visits compared to TCs (Figure 1). Only 4.3% (*n* = 9) of patients would prefer a TC in case of new complaints.

### 3.5. Statistical Analysis

The participation, level of satisfaction, and attitude towards TCs were unrelated to patient’s demographics like age, sex, educational status, years of affiliation to a clinic, or digital fluency (Table 4). The subjects able to use PIA were significantly more willing to contact their doctor via TCs in the future (*p* = 0.011). TCs were rated significantly better by patients who used it only in the nephrology and/or transplantology outpatient clinics compared to patients who also had remote visits at other specialist clinics (*p* = 0.016, R = −2.429).

## 4. Discussion

From March 2020 to the end of 2021, nephrologists and transplantologists from the entire Lodz Voivodeship conducted 31,694 visits, of which 36.5% (*n* = 11 578) were TCs [20]. In this period, there were a total of 16 754 nephrology and/or transplantology consultations conducted in our studied outpatient clinics, of which 38.92% (*n* = 6 520) were TCs.

Although the concept of telemedicine has been known and occasionally implemented since the 1970s [6], TCs were only introduced on a large scale during the COVID-19 pandemic in order to maintain uninterrupted care of patients in both general and specialist care. In Poland, specialist TCs had not been conducted before the pandemic. Lack of any preparation, guidelines, or efficient technological facilities, as well as the need to ensure privacy during remote consultations, constituted a challenge for both patients and doctors.

In Poland, the periods of greatest increases in COVID-19 incidences caused confusion with regard to the form of consultation among patients with chronic diseases, including CKD. Follow-up visits were conducted remotely since, according to the recommendations at the time, only first-time visits, or exacerbation or occurrence of new symptoms required a face-to-face appointment.

Our study showed clearly that patients under long-term, specialist care of a nephrologist or transplantologist prefer face-to-face visits. Despite the positive assessment of TCs during the pandemic—the recommended form of clinical appointment at the peaks of COVID-19 incidences—the patients generally opted for direct contact with their doctor regardless of their distance to a clinic, education level, or experience with remote services in the past. The only group of patients willing to use TCs in the future were those using the Patient Internet Account (PIA). In conclusion, it can be said that for patients in regular, chronic care, personal contact with their doctor is crucial.

As reported in Australian and American studies, for selected groups of patients—especially those living in distant, medically underserved areas—the use of TCs seems to be rated as good as face-to-face visits [24] and can even improve adherence to therapy and treatment outcomes [24]; however, this is on the condition that ambulatory appointments are also accessible when necessary. Interestingly, our study did not show that remoteness affects either satisfaction with or willingness to use TCs in the future. According to previous studies conducted on this subject, TCs are undoubtedly time- and cost-savers [25,26]; however, the numerous limitations of remote care [25] cannot be overlooked. The many positives of TCs in providing remote care highlighted by some researchers may be explained by geographical aspects, e.g., in vast and sparsely populated areas of some countries medical assistance cannot be anything but remote; therefore, remote visits have been long in use there, a fact that has contributed to greater experience and better facilities developed to successfully provide this form of medical help [8,10].

Even with the use of video consultation, which in some way enables inspection of a patient, in addition to a medical interview, doctors cannot perform a complete physical examination. This may lead to a misdiagnosis, delay in diagnosis, or failure to establish any diagnosis, the implication of which could be progression of the disease to an advanced stage, with the cost of its subsequent treatment frequently surpassing the sums saved thanks to TC [27].

Less than half of our respondents stated their preference for remote visits when they only need to obtain repeat prescriptions but face-to-face visits when needing a clinical consultation. In the primary care patient survey report published by the Polish National Health Fund in August 2020 [14], more than 43% of patients marked TCs as the main form of contact, during which the doctor decides whether a face-to-face visit is necessary. The greatest advantages indicated by our patients were a reduced risk of infection and the possibility to contact a specialist without leaving their home. For the majority of the patients, the absence of physical examination was the main disadvantage of TCs. In the assessment of TCs, one-fifth of the patients reported technical issues during the call and/or mentioned problems with the description of their symptoms. The study conducted by George K. et al. obtained the same results with regard to the most common advantages as well as disadvantages of TCs conducted among nephrology patients [28]. Their study, like ours, assessed patients’ satisfaction with TCs expressed on a scale from 1 to 5, and the number of positive ratings (4 or 5) was similar to that obtained in our analysis (90% positive ratings in the Indian study vs. 85.3% in our population) [27].

In our study, we used the Net Promoter Score (NPS), which is an objective loyalty ratio that estimates patients’ experience and satisfaction based on a single survey question, a method that is gaining popularity despite its limitations [29,30]. Although the level of satisfaction with TCs was high in our study, the NPS had a negative score (−14.56), which shows the respondents’ reservation to recommend TCs in specialist outpatient clinics to their relatives/friends. For the Polish primary healthcare system, however, the result of NPS seems to be completely different. In the study conducted by the Polish Ministry of Health, which covered over fourteen thousand patients using TCs in primary care units, the NPS had a positive value of 33 [14]. Such a discrepancy may indicate a decreasing interest in using TCs in specialist medical care, in contrast to primary care. Despite longer waiting times for a visit, patients still prefer to have a face-to-face consultation with a specialist [31].

Our study, like many other surveys, proves that patients were satisfied with specialist TCs [32,33]. However, we observed an interesting fact. Our TC group of patients living over 50 km away from their outpatient clinic was larger than the group of patients living in the same city. It is worth noting that although commuting between their place of residence and the clinic involved more time and money, and exposed them to a greater risk of infection, those patients overwhelmingly stated their preference for face-to-face appointments over TCs.

Just as important as the patients’ opinion of remote visits is the opinion of physicians providing this type of advice. Previous studies have shown the numerous limitations of TCs that might interfere with establishing a correct diagnosis but, on the other hand, convey a belief in telemedicine as a tool to increase the patient’s compliance and/or the effectiveness of therapeutic intervention [12]. A systematic review of 37 studies assessing physicians’ satisfaction with telehealth classified nearly 90% of the studies as those in which the satisfaction level with telemedicine was moderate to high [34]. The high level of satisfaction with telemedicine of both patients and doctors leads us to believe that TCs will be complementary to traditional visits regardless of the epidemiological situation in the future.

With telephone consultations, the good quality of connection; lack of disruptions; and, in the case of videoconferences, a clear, undistorted picture are crucial factors for intelligible communication between patients and doctors, thus ensuring higher adherence to the doctor’s recommendations and more satisfaction with the care provided. According to an American study conducted by Orrange et al., a videoconference, despite the possibility of visual inspection of the patient, is connected with a greater risk of disturbance than a phone call [35]. In order to raise the quality of telehealth services, healthcare organizations should continue developing efficient digital technologies and strategies that improve the use of TCs. There is also another group of patients to be taken into consideration, i.e., elderly people who represent an increasing proportion of society [36] and who may require assistance with technological operating systems. In our study, 5.7% (*n* = 12) of the TC group reported the necessity of assistance—those were not only seniors, but also two patients aged 21 and 39 with a mental disability.

## 5. Limitations

Although the study was conducted in two centers in one large city and many patients commuted from distant areas, the examined population was not very numerous. The survey was conducted only on the patients belonging to the nephrology or posttransplant outpatient clinics, and the survey did not specify which clinic the patient belonged to, which made any comparison between the two groups of patients impossible.

## 6. Conclusions

In conclusion, most of our surveyed respondents of the nephrology and transplantology outpatient clinics rated TCs positively and were willing to have remote medical counsel in the future. However, having an option, the patients would choose traditional, face-to-face visits to specialist outpatient clinics. The patients’ profile, demographic data, place of residence, and distance to the clinic had no impact on their assessment of TCs. Health organizations should develop technology and strategies to improve TCs but only as a system supplementary to stationary appointments, which remain the gold standard.

## Figures and Tables

**Figure 1 healthcare-11-02737-f001:**
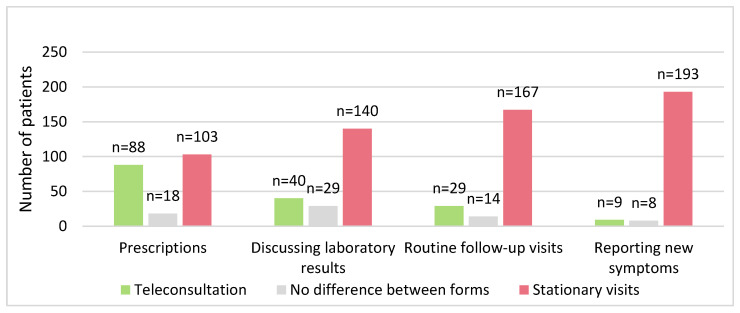
TC group’s preferences towards the form of appointments in the future.

**Table 1 healthcare-11-02737-t001:** Demographic data of all respondents and those who participated in TC (TC group).

Variables	All Respondents (*n* = 294)	TC Group (*n* = 212)
Age (years)	57.06 ± 15.64	55.24 (44–68)
Gender (%)		
male	54.8 (*n* = 161)	53.8 (*n* = 114)
female	43.9 (*n* = 129)	44.8 (*n* = 95)
No response	1.3 (*n* = 4)	1.4 (*n* = 3)
Occupation (%)		
Student	0.3 (*n* = 1)	0.5 (*n* = 1)
Working	35.7 (*n* = 105)	38.7 (*n* = 82)
Unemployed	2.4 (*n* = 7)	2.8 (*n* = 6)
Retired/Pensioner	61.6 (*n* = 181)	58.0 (*n* = 123)
Level of education (%)		
Primary	5.5 (*n* = 16)	5.6 (*n* = 12)
Secondary	67.3 (*n* = 198)	65.6 (*n* = 139)
University	27.2 (*n* = 80)	28.8 (*n* = 61)
Place of residence (%)		
Rural	28.6 (*n* = 84)	30.7 (*n* = 65)
City < 10,000 inhabitants	3.8 (*n* = 11)	4.2 (*n* = 9)
City 10,000–50,000 inhabitants	17.3 (*n* = 51)	18.4 (*n* = 39)
City 50,000–100,000 inhabitants	7.1 (*n* = 21)	5.6 (*n* = 12)
City 100,000–500,000 inhabitants	7.1 (*n* = 21)	8.5 (*n* = 18)
City > 500,000 inhabitants	36.1 (*n* = 106)	32.6 (*n* = 69)

**Table 2 healthcare-11-02737-t002:** Characteristics of TC group.

Variables	TC Group (*n* = 212)
Belonging to Outpatient Clinic (%)	
<6 months	3.3 (*n* = 7)
6 months–2 years	11.8 (*n* = 25)
2–5 years	20.8 (*n* = 44)
>5 years	59.0 (*n* = 125)
No response	5.1 (*n* = 11)
Distance from Outpatient Clinic (%)	
Same city	29.7 (*n* = 63)
<10 km	3.8 (*n* = 8)
10–50 km	26.9 (*n* = 57)
>50 km	37.3 (*n* = 79)
No response	2.3 (*n* = 5)
Number of in-person visits to Outpatient Clinic (%)	
0	4.7 (*n* = 10)
1	5.2 (*n* = 11)
2–4	27.4 (*n* = 58)
5 or more	57.1 (*n* = 121)
No response	5.6 (*n* = 12)
Remote visit to another specialist outpatient clinic (%)	
No	35.4 (*n* = 75)
1	20.3 (*n* = 43)
2 or more	40.6 (*n* = 86)
No response	3.7 (*n* = 8)
Number of in-person visits to family clinic (%)	
0	23.1 (*n* = 49)
1	9.9 (*n* = 21)
2–4	36.8 (*n* = 78)
5 or more	29.7 (*n* = 63)
No response	(*n* = 0)
Number of in-person visits to family clinic (%)	
0	9.9 (*n* = 21)
1	14.2 (*n* = 30)
2–4	42.5 (*n* = 90)
5 or more	29.7 (*n* = 63)
No response	3.7 (*n* = 8)
Use of remote services (%)	
Yes	65.1 (*n* = 138)
No	34.0 (*n* = 72)
No response	0.9 (*n* = 2)
Use of Patient Internet Account (%)	
Yes	51.4 (*n* = 109)
No	48.1 (*n* = 102)
No response	0.5 (*n* = 1)

**Table 3 healthcare-11-02737-t003:** Advantages and disadvantages of TC visits reported by their users.

Advantages	TC Group (*n* = 212)	Disadvantages	TC Group (*n* = 212)
Not leaving home (%)	61.3 (*n* = 130)	No physical examination (%)	59.0 (*n* = 125)
Lower infection risk (%)	54.2 (*n* = 115)	Higher risk of misdiagnosis (%)	32.5 (*n* = 69)
Shorter waiting time (%)	35.8 (*n* = 76)	Problems with describing symptoms (%)	19.3 (*n* = 41)
No benefits (%)	9.9 (*n* = 21)	Technical problems (%)	19.9 (*n* = 38)
Other (%)	0.9 (*n* = 2)	Not understanding doctors’ orders (%)	4.7 (*n* = 10)
		No disadvantages (%)	22.6 (*n* = 48)
		Other (%)	0.9 (*n* = 2)

**Table 4 healthcare-11-02737-t004:** Statistical analysis of factors determining positive assessment and willingness to have TCs in the future.

Variables	Telemedicine	
	Positive Assessment	Future Participation
Sex (M/F)	*p* = 0.326	*p* = 0.327
Use of remote services (Yes/No)	*p* = 0.351	*p* = 0.128
Use of Patient Internet Account (Yes/No)	*p* = 0.233	*p* = 0.011
Participations in TCs in primary health care (Yes/No)	*p* = 0.184	*p* = 0.373
Time of affiliation to a clinic (0–5)	*p* = 0.056 R = 0.138	*p* = 0.110 R = 0.114
Distance from clinic (0–3)	*p* = 0.078 R = 0.126	*p* = 0.113 R = −0.111
Number of TCs in studied clinics (0–2)	*p* = 0.092 R = 0.125	*p* = 0.147 R = −0.105
Participation in TCs in other specialist clinics (0–2)	*p* = 0.016 R = −2.429	*p* = 0.802 R = 0.018
Age	*p* = 0.638 R = −0.334	*p* = 0.098 R = 0.115
Place of residence (0–5)	*p* = 0.334 R = −0.069	*p* = 0.077 R = 0.122
Employment (0–4)	*p* = 0.579 R = 0.339	*p* = 0.716 R = 0.023
Level of education (0–3)	*p* = 0.339 R = 0.068	*p* = 0.234 R = −0.083

## Data Availability

Data sharing is not applicable to this article.

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
