# Peer review of "Assessments of and Attitudes towards Specialist Teleconsultations among Nephrology and Posttransplant Outpatients during the COVID-19 Pandemic"

_healthcare, 2023, doi:10.3390/healthcare11202737_

Round 1

Reviewer 1 Report

Thank you for the opportunity to review this paper. Whilst not an original piece of research, this is original in Poland and will inform their telehealth practices. I have no critical feedback other than the discussion could be strengthened through the discussion about the implications for policy in Poland (practice is well addressed) and further research opportunities.

1. What is the main question addressed by the research? It was examining how telehealth was being used in the outpatient setting

2. Do you consider the topic original or relevant in the field? Does it address a specific gap in the field? Whilst this is not original research with regards to telehealth, it was original in Poland

3. What does it add to the subject area compared with other published material? It does for Poland. Given each health ecosystem is different and health policy is different, it is entirely appropriate to view this as original work that will inform policy and practice in Poland

4. What specific improvements should the authors consider regarding the methodology? What further controls should be considered? None, it was a well designed study and reported in a way that is reproducible. I could not fault it

5. Are the conclusions consistent with the evidence and arguments presented and do they address the main question posed? Yes

6. Are the references appropriate? Yes

7. Please include any additional comments on the tables and figures. None

Author Response

Thank You for Your positive review.

Reviewer 2 Report

Dear Authors,

I have completed my review of your manuscript titled "The Assessment of and Attitude Towards Specialist Teleconsultations Among Patients of Nephrology and Posttransplant Outpatient Clinics in the COVID-19 Pandemic." Your work addresses an important and timely topic, examining the quality and patient satisfaction of teleconsultations among high-risk nephrology and kidney transplant outpatients.

While the manuscript is engaging and generally well-written, there are several areas where it could be further strengthened. Below are my suggestions:

1. Sample Size Justification: The study includes 294 adult patients, but the manuscript does not elaborate on why this sample size is considered adequate for drawing meaningful conclusions. A more detailed discussion about the representativeness of the sample would add robustness to the study.

2. Addressing Preferences for In-Person Consultations: While your results are generally positive concerning teleconsultations, the manuscript would benefit from an exploration of why some patients still prefer face-to-face visits. Understanding the limitations of teleconsultations from the patient's perspective could provide valuable insights for future improvements in this delivery method.

3. Comparative Analysis with Prior Research: The study would benefit from a more comprehensive review of existing research on the topic. Even though your study found that patients' profiles and demographics did not significantly impact their assessment of teleconsultations, comparing these results with other research could provide additional context. For instance, how do your findings compare with studies that consider different patient demographics or types of consultations?

4. Physician Satisfaction Metrics: The manuscript indicates that 94.8% of teleconsultations "fulfilled the purpose of the consultation," but it's unclear how this aligns with physician perspectives. Do physicians share the same level of satisfaction, or are there discrepancies? Addressing this could provide a more rounded view of the efficacy of teleconsultations.

5. Balanced Discussion of Advantages and Disadvantages: Your paper tends to focus more on the advantages of teleconsultations, offering a somewhat imbalanced perspective. A more comprehensive discussion that includes both the pros and cons would enrich the manuscript. Moreover, it would be useful to offer specific recommendations based on your findings, such as how healthcare providers could optimize the teleconsultation experience for patients in these specialty areas.

Thank you for your contribution to our field of study. I look forward to your revised manuscript.

Author Response

We kindly thank you for your suggestions. Below you will find responses to specific comments.

  1. We consider the group to be representative, as it included patients from the only 2 transplant centres in the region, and the study included almost all (response rate was 89.9%) patients who had visits during the study period - there was no selection of patients included in the study. 
  2. To find out individual preferences would require analyzing each survey individually. We did not analyze the preferences of individual patients, but of the entire group.  Based on the survey-based research, we can only presume that the disadvantages of teleconsultation indicated by patients, as well as the advantages of traditional visits, determine the preference for face-to-face visits. 
  3. We have tried to extend the discussion slightly with comparing results to newer study. However, it is not easy to compare, as there are few such studies, plus each country or region differs in its organization and experience in conducting TCs. 
  4. Thank you for your comments, our survey tested the perception of remote visits from the patient's perspective only, but perhaps in the future we will expand it to include clinicians.
  5. As the survey showed, the reception of remote advice was positive hence one can get the impression on the preponderance of advantages, but we tried to present both sides of which we indicated e.g. an unfavourable NPS or dominance of stationary visits.We have marked all changes in the manuscript in red.

Reviewer 3 Report

The authors assess the teleconsultation among nephrology patients during the covid19 pandemic. Following are the comments that must be incorporated in the revised version of the paper. English language must be improved. Proofreading by a native English speaker is required.

1.      What is the difference between telemedicine and teleconsultation? In the introduction section, the previous use of teleconsultation is not described.

2.     A poor literature review is provided about teleconsultation in healthcare and how it helped the patients at home. A more detailed literature review and identification of critical factors is required.

3.     What are the significance and impact of your analysis? Summarize the limitations of existing studies and how your study will address these shortcomings.

4.     Line 99: How is the validation done?

5.     Section 2.2: The questionnaire used in this study should be added in this section.

6.     Table 2: What is the meaning of stationary visit? Is it in-person visits?

7.     Line 139: Apart from follow-up appointments (n=130), what were other TCs? Were these TCs new patients?

8.     Lines 147-154: These sentences are unclear. The authors provide a list of advantages and disadvantages, or it was part of an open question.

9.     Table 3: “No benefits” is an advantage? What are the other advantages?

10.  How the disadvantages or shortcomings of the TCs can be improved? Do patients answer this type of question?

11.  Line 156: What does positively mean?

12.  No caption of Figure 1.

13.  Figure 1: Terminology should be consistent. Stationary visits or traditional visits. What is the meaning of “No difference between forms”?

14.  From Figure 1: It looks like patients are using teleconsulting and doing in-person visits during the study period. Traditional visits are more than teleconsultation. So, the study results will be confusing in favor of teleconsultation or in-person visits. This is the most important point that authors should consider in drawing their conclusions. Add more patients and make three groups: traditional visits, a mixture of visits, and TCs only.  

15.  Line 169: Is it Table 3?

16.  Explain Table 4 in more detail. Most of the factors are not related. Relate your conclusions with Table 4 and show how you draw these conclusions based on the statistics.

17.  The study sample is small, and it isn't easy to draw statistically significant conclusions.

18.  Most of the TCs are supported by in-person visits. So, the subjects' confidence in TCs will also depend on the time between TCs and in-person visits or the number of TCs before an in-person visit. The authors require more substantial statistical evidence in support of their claims.

English language must be improved.

Author Response

We sincerely thank you for all your comments and suggestions for improving the manuscript. Below you will find responses to specific comments.

  1. We added definition of teleconsultations in the introduction.
  2.  There are few reliable studies that assess this, especially for specialized consultations. We have tried to extend the discussion as recommended.
  3.  We have included limitations in the manuscript, we have also tried to extend the discussion slightly. 
  4. As we mentioned, the survey was created and revised by specialists working at the clinics conducting the survey. We then conducted the survey on a smaller group of patients under the care of specialized clinics. Each time, we discussed with the patients what they found incomprehensible and incorporated any comments in the final version of the survey.
  5.  According to the guidelines, the questionnaire was added in supplementary section during submission. 
  6. Yes, phrase “stationary” meant in-person/face-to-face visit, we changed this expression in Table 2 to be understood. 
  7. We surveyed only patients who belonged to the mentioned Outpatient Clinics before the pandemic. “Routine follow-up appointment” in subsection 3.3 titled Course of TCs meant periodic visits (for example, when a patient comes for a check-up every 6 months). Other TCs were due to, e.g. new complaints or get prescriptions. 
  8. Patients selected advantages and disadvantages from a list, had the option to select multiple options, and could indicate their own pros and cons in the open part of the question. 
  9. Thank you, of course, the absence of benefits is not an advantage, but we wanted to show in the table how many patients did not indicate any advantage or disadvantage. 
  10. Unfortunately, our questionnaire did not include such a question. 
  11. Positively in L-156 mean a rating of 5 or 4 on a scale of 1 to 5. We added this information in the manuscript. 
  12. Thank you very much for your notice. We added correct caption, which lost during editing.
  13. We have corrected an unfortunate phrase “traditional”. “No difference between forms” means that the patient does not see the advantage of any form of consultation.
  14. Unfortunately, such separation of patients is impossible. Every patient had inpatient visits before the pandemic, and our survey was designed to examine patients' views on remote visits, which was the only form of contact with a doctor during the peak of Covid-19 cases. 
  15. We edited the mentioned line.
  16. The lack of correlation is also a conclusion that most factors do not affect this group's perception of TCs. We based our conclusions on the results of statistical analysis.
  17. We consider the group to be representative, as it included patients from the only 2 transplant centres in the region, and the study included almost all (response rate was 89.9%) patients who had visits during the study period - there was no selection of patients included in the study. 
  18. Regrettably, it would be very hard to evaluate, some patients have had remote visits multiple times.

We have marked all changes in the manuscript in red.

Round 2

Reviewer 3 Report

The authors have answered my comments

English language must be improved.